



# Novel extensions to the Fisher copula to model flood spatial dependence over North America

Duy Anh Alexandre[1], Chiranjib Chaudhuri[1], and Jasmin Gill-Fortin[1]

[1]Geosapiens Inc., Quebec, QC Canada

**Correspondence:** Duy Anh Alexandre (duy-anh.alexandre@geosapiens.ca)

**Abstract.** Taking into account the spatial dependence of floods is essential for an accurate assessment of fluvial flood risk. We propose novel extensions to the Fisher copula to statistically model the spatial structure of observed historical flood record data across North America. These include a machine-learning based XGBoost model, exploiting the information contained in 130 catchment specific covariates to predict discharge Kendall's $\tau$ coefficients between pairs of gauged–ungauged catchments. A novel conditional simulation strategy is utilized to simulate coherent flooding at all catchments efficiently. After subdividing North America into 14 hydrological regions and 1.8 million catchments, applying our methodology allows to obtain synthetic flood event sets with spatial dependence, magnitudes and frequency resembling those of the historical events. The different components of the model are validated using several measures of dependence and extremal dependence to compare the observed and simulated events. The obtained event set is further analyzed and supports the conclusions from a reference paper in flood spatial modeling. We find a non-trivial relationship between the spatial extent of a flood event and its peak magnitude.

## 1 Introduction

Flooding is one of the most frequent and damaging class of natural disaster, accounting for 43 % of all major recorded events and affecting nearly 2.5 billion people worldwide during the period 1994–2013 (Centre for research on the epidemiology of disasters, 2015). In hydrology, accurate predictions of the frequency and magnitude of extreme flood events are needed to prevent and mitigate the physical and financial damage caused by flooding.

In fluvial flood models, extreme river discharges are usually derived using observed gauge data. They are commonly used as input to hydraulic models to produce fluvial flood depth maps. Traditionally, flood studies are conducted locally, on the scale of a river basin. Many local studies would then be combined to produce flood maps on a national or continental scale. This way of proceeding comes with its limits. Since the methodologies used to assess flood risk can significantly differ from study to study, the aggregation of numerous local studies often lacks coherence (Bates et al., 2023). With current advances in hydrological science, big data and computing capabilities, flood mapping at a national and continental scale is now possible, taking advantage of the power of more data input (Nearing et al., 2021; Wing et al., 2020; Bates et al., 2021).

Fluvial flooding is inherently a spatial phenomenon: large-scale precipitation and the connectivity of river networks mean that extreme discharge can be present simultaneously at different locations. Modeling widespread flooding is of particular interest for the insurance and reinsurance sectors, since they often operate at a regional and national scale. Some recent studies



have shed light on the spatial structure of riverine flood (Brunner et al., 2020; Berghuijs et al., 2019), but this topic is largely understudied. The majority of flood frequency analysis studies does not consider flood spatial dependence, assuming that the same return period flood affects a whole area, an assumption which breaks down if the modeled area is large (Quinn et al., 2019).

Only a few studies have tackled modeling the spatial structure of flood events, usually under stationary conditions (Brunner et al., 2019; Diederen et al., 2019; Keef et al., 2009; Neal et al., 2013; Quinn et al., 2019). Statistical approaches are commonly used to model multivariate extremes, using the theory of max-stable processes (de Haan and Ferreira, 2006; Davison et al., 2012) or copulas (Genest and Favre, 2007). Max-stable processes are parametric models which generalize the univariate extreme value theory to more than one dimension. They have a solid theoretical basis, but because there is no simple parametric

form for the target distribution, statistical inference is significantly harder when the number of dimensions is large. Copulas are statistical multivariate structures which separate the marginal distribution of variables from their dependence structure (Genest and Favre, 2007). Copulas have regularly been used to model multivariate extremes in environmental and climate studies: the extreme value copulas (Lee and Joe, 2018; Ribatet and Sedki, 2013), the Archimedean family of copulas (Schulte and Schumann, 2015; Alaya et al., 2018), spatial vine copulas (Gräler, 2014). Brunner et al. (2019) use the Fisher copula to

model co-occurrent extreme floods in a small river basin in Switzerland, with a proposed simple extrapolation to ungauged catchments. Quinn et al. (2019) use the Heffernan & Tawn exceedance model to analyze the spatial dependence between flood peaks for the contiguous United States. They found that extreme riverflows were positively correlated, with the correlation decreasing as the distance separating two stations increases. The Heffernan & Tawn conditional exceedance model is gaining popularity in spatial flood modeling, thanks to its ground in extreme value theory, its flexibility and easy implementation com-

pared to max-stable processes (Heffernan and Tawn, 2004; Keef et al., 2009). However, compared to copulas, the conditional exceedance model needs the fitting of many regression models, and does not present a natural framework to assess spatial dependence at ungauged catchments (Brunner et al., 2019).

In order to create a flood map with complete spatial coverage, methods to model flows in ungauged catchments are needed. Very few past work have tried to model spatially coherent flood including ungauged catchments. Wing et al. (2020) use

synthetic discharge simulated by a hydrological model to compensate for lacking observed data and study the spatial structure of flood, although computing rainfall-runoff outputs for every ungauged catchment on a continental scale can represent a high computational burden. Brunner et al. (2019) derived a simple method to extend the Fisher copula model to ungauged catchments, where the pairwise Kendall's $\tau$ coefficients are regressed against river distance. The authors tested this method on a small subset of 22 stations in a Swiss river basin.

We propose in this study a methodology to model the spatial dependence of riverine flood on a continental scale, based on the Fisher copula model developed in Brunner et al. (2019) and Favre et al. (2018). This model is extended to allow for efficient simulation of floods with spatial coherence in ungauged catchments. The interpolation method in Brunner et al. (2019) is modified to calculate spatially coherent flows for every catchment, using machine learning and a new conditional simulation strategy. Using this probabilistic approach, a synthetic catalogue of stochastic discharge events resembling the observed record





can be produced, with plausible discharge magnitudes not previously seen. The discharge event set can subsequently be used
in hydraulic models to produce a flood depth event set and assess corresponding physical damage and financial losses.

Our methodology includes the following components, which will be addressed in the following sections:

– Event identification from observed records of daily discharge and creation of a regional observed event set, section 3.1.

– Dependence model for flood events based on the Fisher copula, section 3.2.

– Interpolation of the Fisher copula model to ungauged catchments, section 4.1.

– Simulation of a synthetic event set for all catchments, with spatial dependence, section 4.2.

– Event timestamping (frequency component) and back-transformation with a marginal model for extreme discharge (magnitude component), section 4.3.

## 2  Data

### 2.1  Daily discharge data

We use daily discharge data from various sources, first starting with data from The Global Runoff Data Centre database (GRDC, The Global Runoff Data Centre), an international data center operating under the auspices of the World Meteorological Organization. It consists of 2118 stations high quality records across North America, with different time periods for each station. This dataset is supplemented by various government sources: USGS National Water data for the USA (U.S. Geological Survey, 2016), Atlas hydroclimatique for Quebec (Ministère de l'Environnement, de la Lutte contre les changements climatiques, de
la Faune et des Parcs, 2022) and HYDAT Database for Canada (Environment Canada, 2013), extracted with the TidyHydat package in R. Because some bad quality data was found in the latter data sets, a quality control procedure is devised to select only the station records considered reliable. For this end, we compare annual maximum discharges calculated from the daily discharge series with those from a separate annual maxima database. Annual maximum instantaneous peak discharge datasets
are extracted directly from USGS National Water data for the USA (U.S. Geological Survey, 2016) and HYDAT Database for Canada (Environment Canada, 2013). The annual maxima dataset covers 23283 stations with at least 10 years of quality-checked data. Mean relative error is calculated across concurrent annual maxima for each station. Stations with a mean relative error lower than 10 % calculated over at least 10 years are considered reliable and added to the GRDC data.

To focus on stations with significant discharge, only those with a drainage area higher than 50 km$^2$ are kept, and any
station–year with non-complete daily data is discarded. After this preprocessing, 3385 high quality stations are used over North America.

### 2.2  Catchment attributes

A large dataset of 130 characteristics for over 1.8 million catchments across North America is used to model floods at ungauged catchments. These include catchment-specific attributes in term of geography (latitude, longitude, altitude), hydrology (average





stream slope, drainage length, drainage area, flow direction fraction), climate (solar radiation, wind speed, vapor pressure) and soil properties (silt, sand, clay and gravel fraction of different soil layers, mean base saturation and fractional coverage of different land coverage classes). For the detailed description of each variable as well as their sources, the reader can refer to Table S1 in the supplementary information.

## 2.3  Catchment delineation

In this section, we describe the methodology employed to generate the river network, delineate catchment boundaries, and establish topological relationships between catchments. The process utilizes a 90 m spatial resolution MERIT-HYDRO Digital Elevation Model (DEM) with an average unit catchment size of 10 km$^2$ (Yamazaki et al., 2019). The 90 m MERIT-HYDRO DEM forms the basis for deriving the flow direction raster, which is then used to compute the flow accumulation raster. Watershed delineation is performed for each basin, employing a threshold of 10 km$^2$. The delineation process involves specific

functions within the Arc Hydro Toolbox, following a classic watershed delineation approach, namely: StreamDefinition, Stream Segmentation, CatchmentGridDelineation, CatchmentPolygonProcessing, and DrainageLineProcessing.

This methodology is consistently applied across all major watersheds identified within the 90 m MERIT-HYDRO dataset. The primary objective is to optimize watershed sizes to an approximate average of 10 km$^2$, contrasting with the typical dataset norm of around 100 km$^2$. The integration of functions within the Arc Hydro Toolbox ensures precise attribution of each line to

its respective catchment, effectively delineating its connections with downstream features. The scope of this study encompasses the entirety of North America, resulting in a comprehensive data set comprising 14246 network groups (NetID) and 1821571 delineated catchments (COMID).

Subsequently, each discharge station is mapped to a combination of NetID and COMID, providing a comprehensive association between the stations and their respective catchments within the river network. The decision was made to associate

each station with a catchment to simplify the computation of catchment attributes and ensure accurate alignment with the corresponding watercourse. Discrepancies between the actual station position and the reported position are not uncommon. For each station, the disparity between the reported drainage area and the cumulative upstream drainage area of the surrounding basins is calculated. The station is then linked to the watershed with the smallest error. Subsequently, only stations with a difference between the station's drainage area and the catchment area falling below the maximum of either 15% of the upstream

catchment drainage area or 60% of the catchment unit area (addressing cases of head catchments), were retained (see Fig. S2, supplementary information for a visual plot of this step).

The catchments are partitioned into 14 hydrologically similar regions across North America, corresponding to level 2 of the HydroBASINS product, which delineates watershed boundaries and provides a seamless global coverage of consistently sized and hierarchically nested sub-basins at different scales (Lehner and Grill, 2013). These regions are numbered and named, in

order: 1–Arctic, 2–Northwest passage, 3–Hudson bay, 4–Nunavut, 5–North-western territories, 6–Alaska, 7–Saint-Lawrence, 8–Prairie, 9–British Columbia, 10–East Coast, 11–Midwest, 12–California, 13–Mexico and 14–Carribean. They are represented in Fig. 1, along with the locations of the preprocessed station data. An example of river network (NetID) and catchment (COMID) delineation is shown in Fig. 2, for region 9, British Columbia.



**Figure 1.** The 14 HydroBASINS regions and location of station data after preprocessing (black dots).

## 3    Dependence model for gauge stations

The subsequent methodology is applied for each HydroBASINS region independently. Catchments within a region can share a same large-scale hydrological behavior, but stations across multiple regions usually exhibit distinct spatial structures. This is supported by recent studies showing that the distance separating catchments experiencing the same floods exhibits regional differences, depending on the nature of flood generating mechanisms (Berghuijs et al., 2019; Brunner et al., 2020). Because of station sparsity, we omit analysis on regions number 1, 2, 3, 4, 13 and 14 (respectively the Arctic, Northwest passage, Hudson

bay, Nunavut, Mexico and the Carribean). In what follows, the methodology and results are presented for region 9–British Columbia. Results with regard to regions 7 (St Lawrence), 8 (Prairie), 10 (East Coast) and 11 (Midwest) can be found in the supplementary information (Fig. S5-S11).





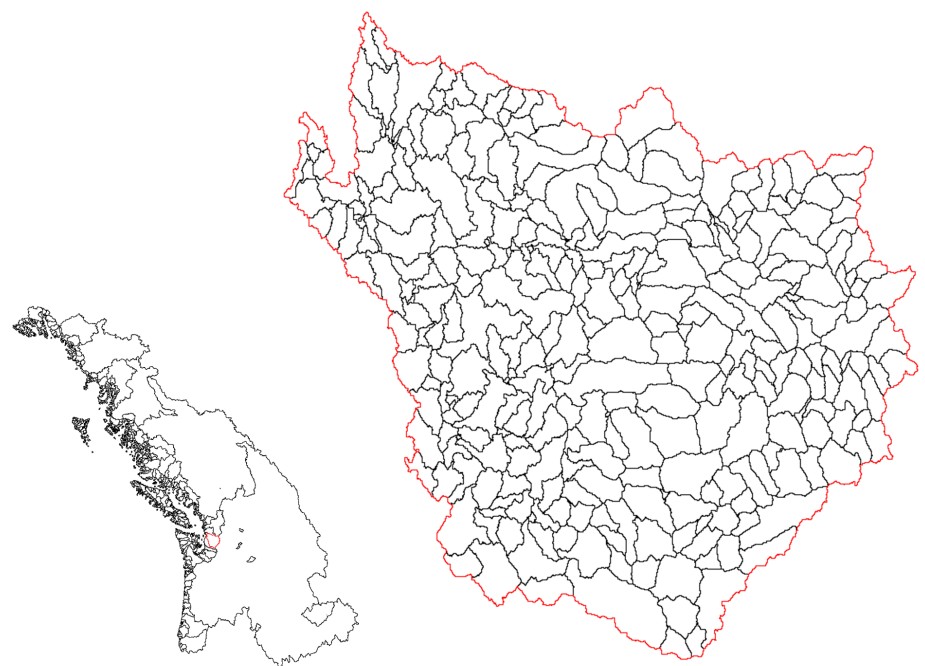

**Figure 2.** Discretization of the HydroBASINS region 9 into river networks (left) and of a small river network into catchments (right). The considered network is highlighted with a red contour.

## 3.1 Event definition

### 3.1.1 Synchronous dataset creation

To capture the spatial structure between discharges from multiple stations, observed data have to be available synchronously. For each region, since station data have variable timespans, we choose a sufficiently long common period of observations which maximizes the spatial coverage. For this end, the 30 years (not necessarily continuous) with the highest number of stations are selected as the common time period. Then only stations with complete data coverage for these 30 years and presenting a negligible trend as detected by the Mann-Kendall trend test are retained for analysis. There is a trade-off to be made between

the time period length and the spatial coverage, as increasing the time period decreases the number of stations with available data. We consider 30 years as enough temporal coverage to capture the extreme discharges, and imposing a longer time period would lead to discarding too many stations for some already sparse regions. For British Columbia, 30 years of discharge record in the period 1975–2013 from 253 stations are used (see Fig. 3a). Gauge density is better in the southern part of the region. The synchronous discharge dataset is then used to define regional events for spatial structure assessment.




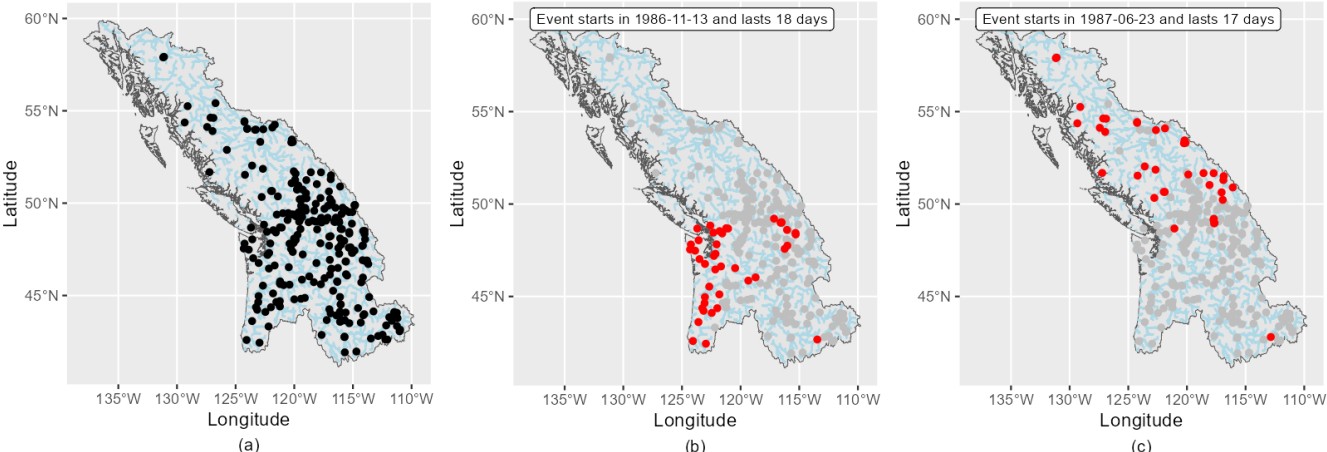

**Figure 3.** Gauge coverage for British Columbia (a) and examples of observed regional event footprints (b, c). Red dots are gauge sites with flows greater than the 0.9[th] quantile flow during a given event.

### 3.1.2 Event identification

Defining what constitutes a flood event in a continuous series of discharge data is a challenging task implying a number of arbitrary choices. First, local flood events are identified as high peaks at each station, using a peak identification algorithm described in Diederen et al. (2019). The algorithm identifies successive alternation of local maxima and minima in the time series, defined as the points where the sign of the increment changes from positive to negative and vice versa. It also includes noise removal and declustering options to remove small perturbations on the one hand, and to ensure events are sufficiently separated temporally on the other hand. The threshold for noise removal is taken as $\delta_y = 0.02 \times \max(dY)$ where $dY$ is the series of absolute differences between successive high and low peaks, similar to the work in Diederen et al. (2019). The minimum separating time between two local events is set to 6 days, as to decluster peaks belonging to a same event and ensure identified events are more or less independent. This is in line with a comprehensive analysis of flood wave travel times by Allen et al. (2018), who shows that, globally, flood waves take a median travel time of 6 days to reach their basin terminus. Finally, only peaks exceeding the 0.9[th] quantile flow for each station are retained, as we are interested in the extreme discharge. Local events are assigned the dates when the maximum peaks occur.

Regional events are identified from the local events, using the 1D mean shift clustering algorithm (Comaniciu and Meer, 2002) to group the dates of all local events into clusters. Each cluster corresponds to a time period that defines a regional event. This definition is able to account for the lag time between peak discharges belonging to a same event. As with any clustering algorithm, a choice has to made with regard to the optimal number of clusters. The algorithm bandwidth is calibrated to obtain between 12 and 24 events per year, on average. This is a trade-off between having enough events to capture the right tail behaviour of discharge, and avoiding considering small fluctuations around real peaks as an event. Events containing only one





local event are discarded, because their natures are not truly regional. A regional event consists in one discharge value for each
station, using the following rule:

- Stations with a local event happening within a regional event duration are assigned its magnitude. In cases where two or
  more local events occur within the regional event, the one with the highest magnitude is used.

- For other stations, the maximum discharge magnitude observed during the regional event is assigned. Those are fill-in
  values, necessary to assess the flood spatial dependence.

Using a data based method to obtain regional events allows for variable event durations, identified from the dates of the local
peaks. This is a more realistic assumption than the approach adopted in Brunner et al. (2019) and Diederen et al. (2019), where
regional events correspond to fixed time windows of 7 and 21 days respectively.

For British Columbia, 588 regional events are identified (an average of 19.6 events per year), with a mean event duration
of 14 days. The mean time interval between two consecutive events is 18.7 days. To describe flood events affecting a large
area, the event footprint is often used, defined as the spatial pattern in flood return period for an event (Rougier, 2013). Two
examples of observed event footprints are shown in Fig. 3b and 3c. Stations shown in red are those exceeding the 0.9$^{\text{th}}$ quantile
threshold. As expected, spatial information is contained in the observed flood events, as high flows tend to occur in spatial
clusters.

## 3.2 Fisher copula model

### 3.2.1 Copula theory

The copula theory is based on the representation theorem of Sklar (1959) which states that any multivariate cumulative distri-
bution function $F_{(X_1, X_2, \ldots, X_n)}$ of a set of continuous random variables can be written as

$$F_{(X_1, X_2, \ldots, X_n)}(x_1, x_2, \ldots, x_n) = P(X_1 \leq x_1, X_2 \leq x_2, \ldots, X_n \leq x_n) \tag{1}$$

$$= C(F_1(x_1), F_2(x_2), \ldots, F_n(x_n)) \tag{2}$$

where $F_1, F_2, \ldots, F_n$ are the marginal univariate distributions and $C$ is termed the copula. One main advantage of the copula
approach is that the dependence modeling can proceed independently of the choice of the marginal distributions (Genest and
Favre, 2007). The Fisher copula, first introduced in Favre et al. (2018), is defined as the dependence structure of the square of
a multivariate Student random vector. It is parametrized by a correlation matrix $\Sigma$ and a degree of freedom $\nu$, corresponding
to the same parameters of the Student's $t$ distribution. When looking at the dependence structure of extreme values, a common
measure of interest is the upper tail dependence coefficient, defined for two variables $X_1$ and $X_2$ with marginal distributions
$F_1$ and $F_2$ as

$$\lambda_U = \lim_{u \to 1} P(X_1 > F_1^{-1}(u) \mid X_2 > F_2^{-1}(u)) \tag{3}$$

This describes the probability that one variable exceeds a high threshold given that the other variable also exceeds a high
threshold (approaching the right tail). The Fisher copula allows for modeling in high dimensions, non vanishing upper tail





dependence and radial asymmetry. It assumes a positive upper tail dependence but asymptotic independence (dependence disappears at very large distances between stations). This is a suitable assumption for riverine flood, as nearby gauges are expected to have correlated discharges, but for distant sites high discharges will almost surely come from independent flood events. When $\nu \to +\infty$, the upper tail dependence vanishes and the Fisher copula approaches the centered chi-square copula.

### 3.2.2 Inference

The Fisher copula parameters are estimated from the regional event set using Kendall's $\tau$ correlations between pairs of stations. The Kendall's $\tau$ correlation between two observed random vectors $\mathbf{x} = (x_1, ..., x_n)$ and $\mathbf{y} = (y_1, ..., y_n)$ is defined as

$$\tau = \frac{2}{n(n-1)} \sum_{i<j} \text{sign}(x_i - x_j)\text{sign}(y_i - y_j) \tag{4}$$

where $\text{sign}(x) = 1$ if $x > 0$ and $-1$ otherwise. It ranges from $-1$ to $1$ and is invariant to one-to-one transformations of the variables, making it relatively independent of the marginal distributions. The two-step pseudo-maximum likelihood estimator and Kendall's $\tau$ inversion method proposed in Favre et al. (2018) are used, exploiting the monotonous relationship between each entry of $\Sigma$ and the pairwise Kendall's $\tau$. Since we work with stations spanning a large spatial scale, some pairs of stations present a negative Kendall's $\tau$ (15 % of the pairs for region British Columbia). Those values are replaced by zero before Kendall's $\tau$ inversion, since it requires the $\tau$ coefficients to be positive. After inversion, the obtained $\Sigma$ matrix is often non-positive definite. We slightly modify the $\Sigma$ matrix by replacing negative eigenvalues in the diagonal matrix by a small value, following the idea of Higham (2002), making it positive definite. Using the estimated Fisher copula, a simulated event set of arbitrary length can be generated for all gauge stations.

## 4 Dependence model extended to ungauged catchments

### 4.1 Kendall's $\tau$ interpolation model

In Brunner et al. (2019), the authors developed a method to extend the Fisher copula model to ungauged catchments, using a regression model to predict the Kendall's $\tau$ between all pairs of sites as a function of their river distances. In this way, the correlation matrix $\Sigma$ is extended to include all catchments, and the new parameters are used to simulate discharge at all catchments. Drawing from this idea, we develop a machine-learning model to take advantage of the vast information contained in the 130 catchment specific characteristics described in section 2.2. Kendall's $\tau$ between each ungauged catchment and gauge station are predicted with a XGBoost model, as it balances out good performance, fast training and scalability. XGBoost is in the family of boosting models, an ensemble machine learning method which sequentially combines several weak leaners (usually shallow decision trees), gradually learning from past mistakes to increase the model performance (Hastie et al., 2009). XGBoost is extensively used by machine learning practitioners to create state of art data science solutions, and it has been used successfully for many winning solutions in machine learning competitions.



The absolute differences between pairwise covariates are used as features in the training data. For better performance and
to reduce the impact of outliers, the 130 covariates are first quantile transformed to normal distributions. Also, because the
Kendall's $\tau$ lie inside $[-1, 1]$, they are logistic-transformed to lie on the real line, using the relationship

$$y = \log\left(\frac{(\tau + 1)/2}{1 - (\tau + 1)/2}\right) \tag{5}$$

then used as the targets. Decision trees are used as the base learner, as this is the common choice and allows non-linearity.
The optimal hyperparameters (number of trees, tree depth, learning rate, data and feature subsampling rates, regularizing
coefficients) are selected by randomized search 10–fold cross-validation on the training data. The Kendall's $\tau$ between pairs of
ungauged catchments are not computed, as they are harder to predict accurately and not needed in the conditional simulation
strategy explained in section 4.2.

### 4.2   Conditional simulation of events in ungauged catchments

The simulated event set for gauge stations is extended to include all ungauged catchments using the conditional simulation
strategy explained below. Let $(\Sigma_g, \nu)$ be the Fisher copula parameters estimated for the gauged catchments.

First, the Fisher copula can be simulated by transforming simulated values from a Gaussian copula with correlation matrix
$\Sigma_g$. If

$$Z_1, \ldots, Z_n \sim \mathcal{N}_n(\mathbf{0}_n, \Sigma_g) \tag{6}$$

$$C \sim \chi^2(\nu) \tag{7}$$

$$X_i = F_{(1,\nu)}(Y_i) \text{ where } Y_i = \frac{Z_i^2}{C/\nu}, \quad i = 1, \ldots, n \tag{8}$$

where $n$ is the number of stations, $\chi^2(\nu)$ is the univariate chi-square distribution with $\nu$ degree of freedom and $F_{(1,\nu)}$ is the
univariate Fisher cumulative distribution function with 1 and $\nu$ degrees of freedom, then $(X_1, \ldots, X_n)$ follows a Fisher copula
and has uniform marginals (Favre et al., 2018). For each simulated event $(X_1, \ldots, X_n)$, the realized value of $(Z_1, \ldots, Z_n)$ can
be stored.

Second, the correlation matrix $\Sigma_g$ is extended to ungauged catchments using the predicted Kendall's $\tau$ from the XGBoost
model and Kendall's $\tau$ inversion. For each ungauged catchment $u$, define an extended correlation matrix $\Sigma_u$ as

$$\Sigma_u = \left[\begin{array}{c|c} \Sigma_g & \mathbf{v}_{g,u} \\ \hline \mathbf{v}_{g,u}^\top & 1 \end{array}\right] \tag{9}$$

where $\mathbf{v}_{g,u}$ are the correlation entries between catchment $u$ and the gauge stations, obtained by inverting the predicted Kendall's
$\tau$ by the XGBoost model described in section 4.1.

Finally, notice that if $(Z_1, \ldots, Z_n, Z_{n+1}) \sim \mathcal{N}_{n+1}(\mathbf{0}_{n+1}, \Sigma_u)$ then $Z_1, \ldots, Z_n \sim \mathcal{N}_n(\mathbf{0}_n, \Sigma_g)$. We can easily simulate vari-
able $Z_{n+1}$, corresponding to catchment $u$, conditionally on realized value $(Z_1, \ldots, Z_n) = (z_1, \ldots, z_n)$, since they are all normal





variables. Computation of unobserved discharge $X_{n+1}$ follows, as described below

$$\mu_u = \mathbf{v}_{g,u}^{\top} \Sigma_g^{-1} \mathbf{z} \tag{10}$$

$$\tau_u^2 = 1 - \mathbf{v}_{g,u}^{\top} \Sigma_g^{-1} \mathbf{v}_{g,u} \tag{11}$$

$$Z_{n+1}|(Z_1 = z_1, ..., Z_n = z_n) \sim \mathcal{N}(\mu_u, \tau_u^2) \tag{12}$$

$$X_{n+1} = F_{(1,\nu)}(Y_{n+1}) \text{ where } Y_{n+1} = \frac{Z_{n+1}^2}{C/\nu} \tag{13}$$

where $C, \nu$ and $F_{(1,\nu)}$ are the same as in Eq. 8 and $\mathbf{z} = (z_1, \ldots, z_n)$. Notice that we use the same value of $\nu$ previously estimated using gauge stations, for reasons discussed in section 7.2. The former procedure is repeated for each ungauged catchment sequentially.

### 4.3   Event set generation

Using the Fisher copula model and proposed extensions to ungauged catchments, a synthetic event set with full spatial coverage can be simulated. Two aspects of the event set need to be addressed at this point, namely the frequency and intensity of simulated events.

#### 4.3.1   Event timestamping

In order to reproduce the temporal distribution of observed events and calculate different annual statistics, simulated events are associated with synthetic dates. Starting from year 0, $N-1$ time intervals separating consecutive events are independently sampled using the empirical distribution of separating time between observed events. $N$ synthetic dates are thus obtained. Each synthetic date is associated with a synthetic event with a given footprint. For historical discharge, events of similar footprint size are usually clustered together in time, representative of seasonal variations between wet and dry periods. To account for
the temporal dependence in event footprint size, we simulate a time series of $N$ event footprints from the empirical distribution, allowing for a positive probability $p$ that footprint size is the same for two consecutive events. Comparison of the observed and simulated correlograms is used to calibrate $p$ in this step. The $N$ desired event footprint sizes are assigned to the $N$ synthetic dates. Finally, events in the simulated set are assigned to the synthetic dates by matching their footprints with the desired footprints, with allowance for some small differences.

#### 4.3.2   Back-transformation of simulated events

Events simulated using the Fisher copula have uniform margins, by definition. A marginal model is needed to back-transform the simulated values to the discharge scale. Since creating the regional discharge data set restricts the amount of available data, a marginal model is developed using the discharge annual maxima dataset described in section 2.1, covering more than 23000 stations. A hierarchical bayesian model is built to estimate the Generalized Extreme Value (GEV) distribution parameters
for annual maximum discharge in all catchments, using the catchment attributes as covariates. Validation of the modeled return period flows against those from government sources can be found in the supplementary information, for region British





Columbia (Fig. S3). Details regarding this marginal model will be the subject of a separate paper, as the focus of this study is rather on the spatial structure of riverine floods.

Directly back-transforming the simulated values would require a marginal model on the event scale. Instead, the marginal
model described above operates on the annual scale. Therefore, before being back-transformed, the simulated values $q$ are scaled using a scaling factor equal to the (observed) mean number of events per year $n_y$:

$$q_{GEV} = 1 - n_y \times (1 - q) \tag{14}$$

Using the scaling factor $n_y$ is broadly similar to converting return levels for exceedances calculated with the Generalized Pareto distribution to yearly return levels, see chapter 4 of Coles et al. (2001). Then, a discharge level which is exceeded in average
once every $T$ years ($q_{GEV} = 1 - 1/T$) is exceeded in average once every $T \times n_y$ events $\left( q = 1 - \frac{1}{T \times n_y} \right)$.

## 5 Model validation

### 5.1 Fisher copula dependence model

The Fisher copula dependence model is validated by comparing the simulated events (for gauge sites) to the observed events, visually and through examining different dependence measures. Figure 4 shows observed and simulated values (on the uniform
margin) for a pair of nearby stations (gauge 1 and 2) and a pair of distant stations (gauge 1 and 3) in region British Columbia. Each point represents a flood event, observed (blue) or simulated (orange). Observed discharges are more correlated between stations 1 and 2 ($\tau = 0.78$) than between stations 1 and 3 ($\tau = 0.55$), reflecting their respective locations, as shown on the leftside map. The Fisher copula is able to suitably reproduce the pairwise correlation of discharge for different dependence strength, especially in the upper tail. This is partly explained by the ability of the Fisher copula to model a wide range of upper
tail dependence values (depending on the values of the $\nu$ and $\Sigma$ parameters), while the lower tail dependence is always 0, see Favre et al. (2018). Therefore, the Fisher copula might be less well suited for variables correlated in the lower tail, although since focus is on extreme discharge, this is unlikely to negatively affect the results.

Several dependence measures are also compared for the observed and simulated events. The Kendall's $\tau$ between all pairs of stations are compared in Fig. 5a, giving a general picture of the dependence structure. For lower values of $\tau$ ($< 0.6$), the
Fisher copula model tends to slightly underestimate the correlation, but this is not the case for values higher than $0.6$. This is partly due to converting a small number of negative Kendall's $\tau$ to zero before inversion of Kendall's $\tau$, which can distort the obtained correlation matrix $\Sigma$. To assess dependence in the extremes, we look at the F-madogram and the upper tail dependence coefficient (UTD). Usually used to validate max-stable processes, the F-madogram is a pairwise statistic summarizing the spatial dependence structure of the data in the upper tail (Cooley et al., 2006), expressed as a function of the distance between
pairs of sites. The F-madogram computed for the simulated events (orange) are compared to the F-madogram of observed events (blue) in Fig. 5b. The dependence pattern is similar for the simulated and observed values. For the UTD, the nonparametric estimator of Schmid and Schmidt (Schmid and Schmidt, 2007) with a cut-off parameter $p = 0.1$ is used to compare simulated and observed samples in Fig. 5c. The boxplot of the UTD differences between simulated and observed events for all pairs





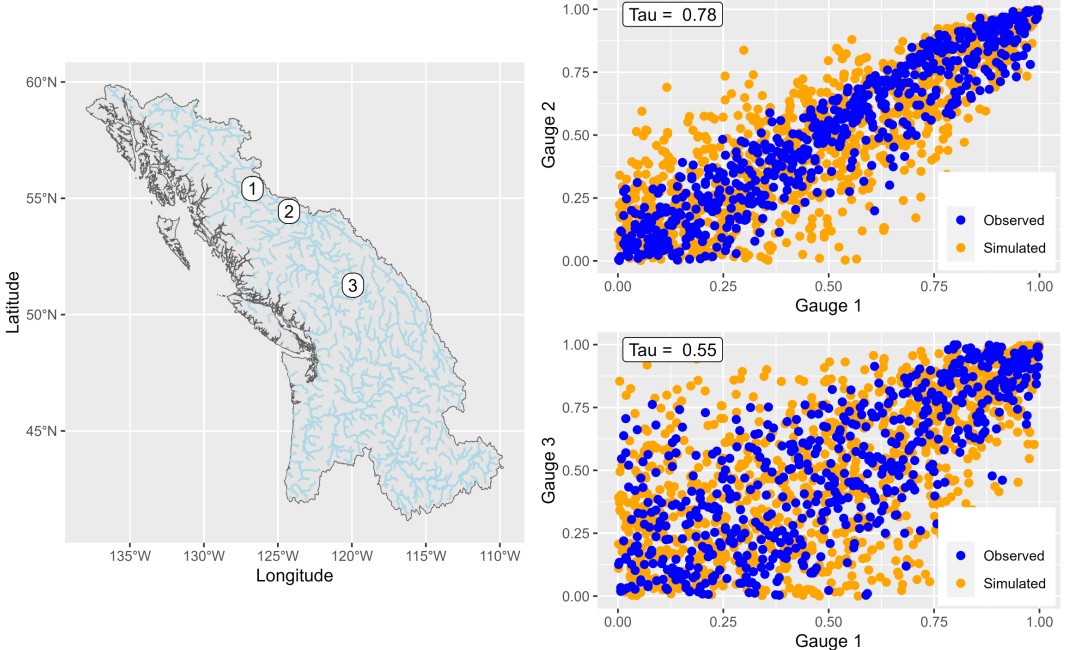

**Figure 4.** Example of pairwise discharge correlation in three stations in British Columbia region. The observed (blue) and simulated (orange) discharge values are shown for pairs of stations 1–2 and 1–3 (right) . The geographical locations of the three gauges are also shown (left).

of stations is shown in Fig. 5d. Upper tail dependence is very slightly overestimated for the British Columbia region. Still,
the comparison shows good agreement, given that this measure is difficult to estimate, especially for limited observed record
length (Serinaldi et al., 2015). Overall, comparison of the different dependence measures shows good agreement between the
observed and simulated event set.

## 5.2   Interpolation model to ungauged catchments

The selected XGBoost model is validated using 10-fold cross validation. This means that 90 % of the training data is used to fit
the model and predictions are made on the remaining 10 % of the data, in sequence. Figure 6a plots out-of-sample predictions
of Kendall's $\tau$ using the cross-validating models against their true values for each pair of stations. The out-of-sample predicting
power of the interpolation model is very satisfying. The Pearson correlation between the out-of-sample predictions and the true
values is 0.97.

    The XGBoost model outputs are plotted spatially to assess the coherence of predicted Kendall's $\tau$ with respect to the gauge
site. One example of this is shown in Fig. 6b. The spatial dependence pattern is well captured by the model here, as the catch-
ments closest to the gauge station exhibit higher correlation, and correlations decrease as the distance increases. Catchments
in the southern part of the region are nearly independent of the gauge site, showing that there is no spatial dependence in
their extreme discharges. Geographical distance does not determine by its own the variability of discharge correlation. After

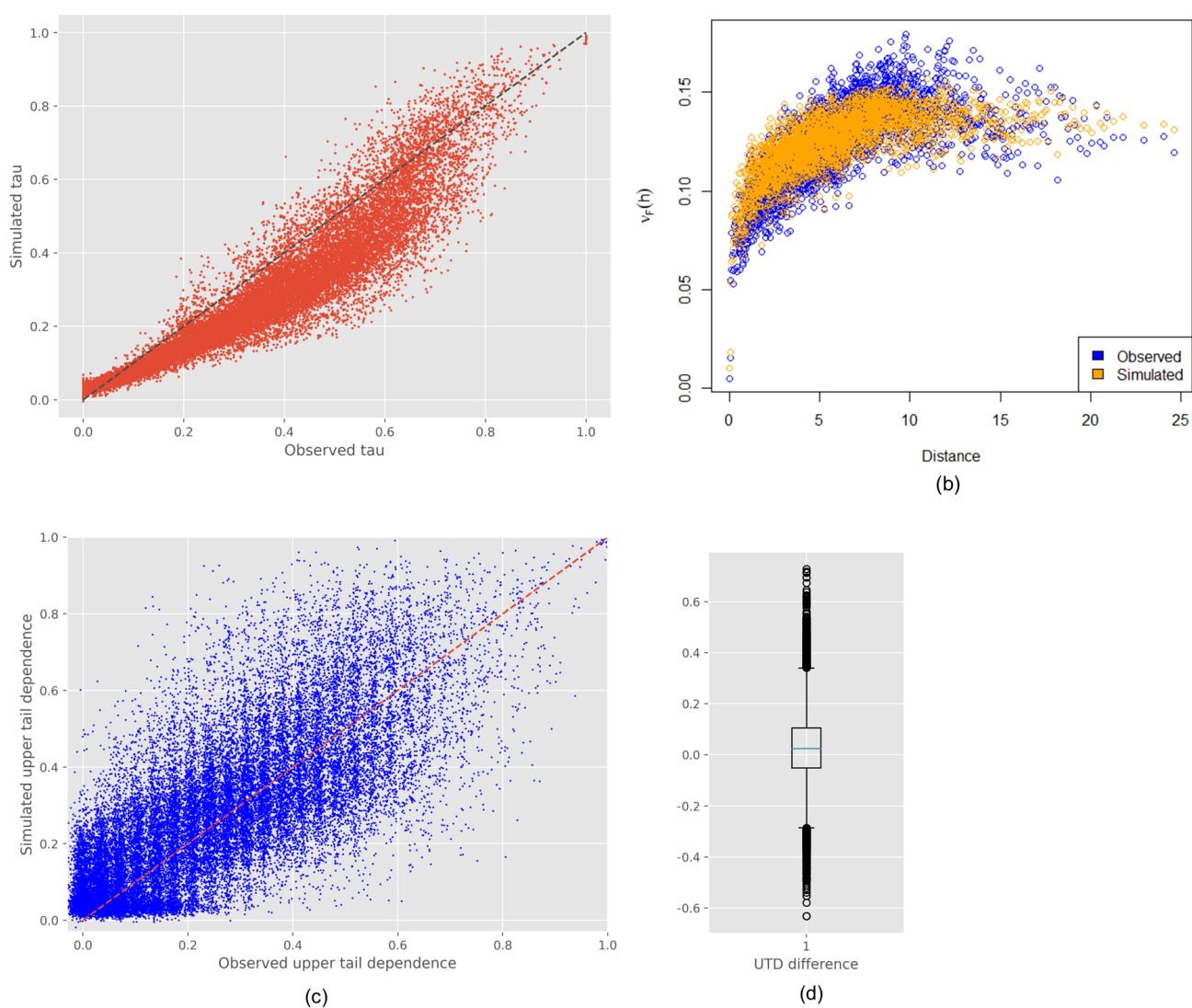

**Figure 5.** Validation of the Fisher copula model is done by comparing different measures for the observed and simulated events: pairwise Kendall's $\tau$ (a), F-madogram (b) and upper tail dependence (c). The boxplot in (d) represents the distribution of the differences between simulated and observed UTDs.



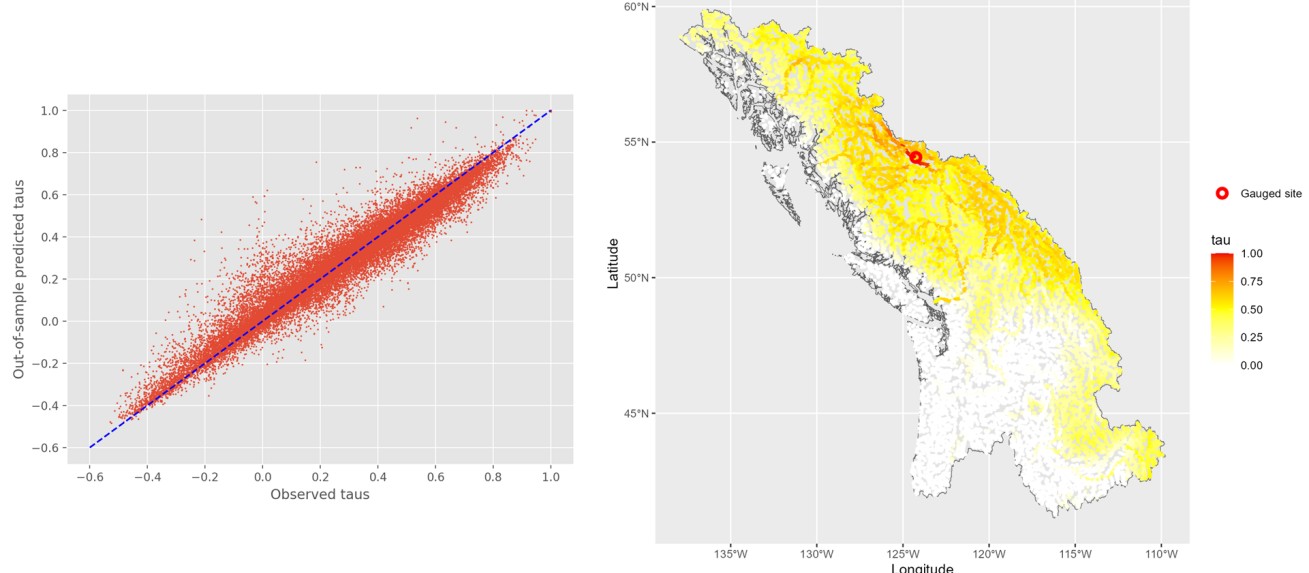

**Figure 6.** Validation of the XGBoost tau interpolation model. Out-of-sample Kendall's $\tau$ predictions are compared with the true values for all pairs of gauge stations (left), using 10-fold cross validation. Predicted values with respect to one gauge site are shown (right).

fitting the model, feature importance in XGBoost can be assessed by counting the number of times each feature is used in
a decision tree. For region British Columbia, the most relevant catchment characteristics are catchment longitude, dam area, solar radiation, catchment latitude followed by wind and some land use indicators (see Fig. S4, supplementary information). The predicted Kendall's $\tau$ also inform on the spatial footprint of events impacting the gauge stations in the data. Some stations measure events with a localized nature (smaller number of predicted high $\tau$), while others are more prone to widespread flood events with a large spatial footprint (not shown).

**5.3  Simulated event set**

To validate the spatial pattern in the simulated event set, it is compared to the observed event set in term of event footprint size, defined as the number of gauge sites exceeding the $0.9^{\text{th}}$ quantile flow (on the regional event scale). Figure 7 shows the normalized histogram of the footprint size for all observed (red) and simulated events (blue). The observed and simulated events have a similar distribution of footprint size, with high footprint events being rarer. Half of the events have a footprint
size lower than 15 sites (over the 253 gauge sites). The largest event in the observed event set impacts 181 gauge sites. Using stochastic simulation, a high number of widespread events can be generated. Similarly to Quinn et al. (2019), we also compare the mean event footprint size at each gauge site, for the observed and simulated events (Fig. 8). The mean is taken over all events impacting a given gauge station with a flow higher than the $0.9^{\text{th}}$ quantile flow. We see that the mean footprint size is reasonably well reproduced in the simulated events, especially for sites with a higher mean event footprint size.



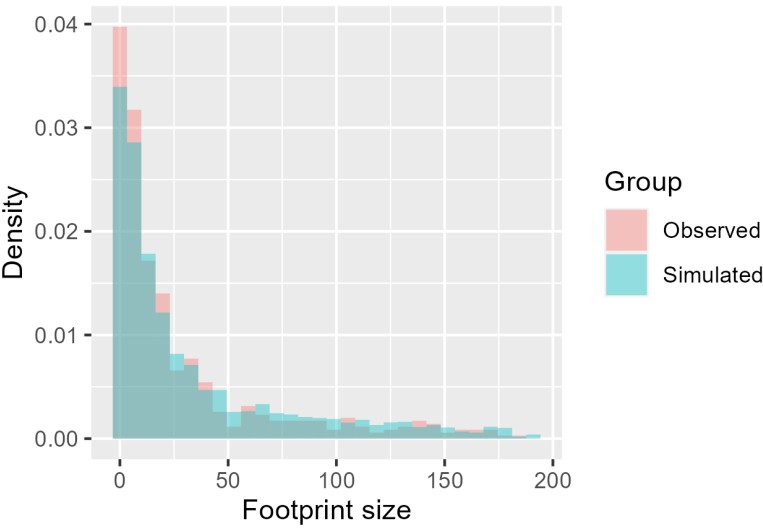

**Figure 7.** Histogram of event footprint sizes, defined as the number of gauges with a flow greater than the $0.9^{th}$ quantile flow, for observed and simulated events.

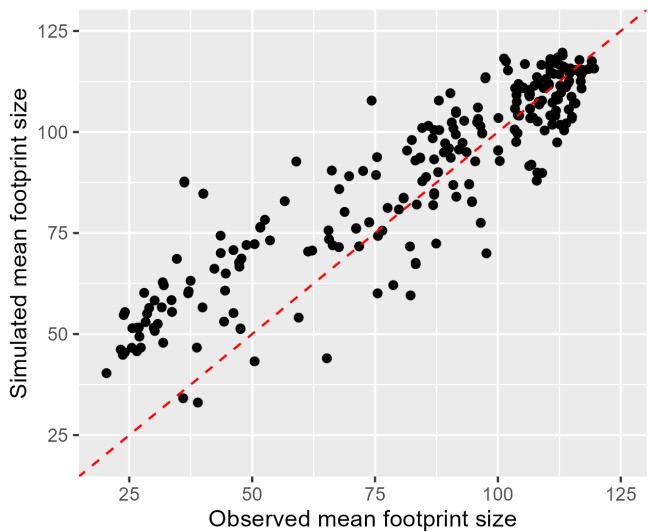

**Figure 8.** Mean event footprint size for the observed and simulated events at each gauge station. The means are calculated over all events impacting each gauge with a flow greater than the $0.9^{th}$ quantile flow. The dotted red line is the diagonal 1:1 line.



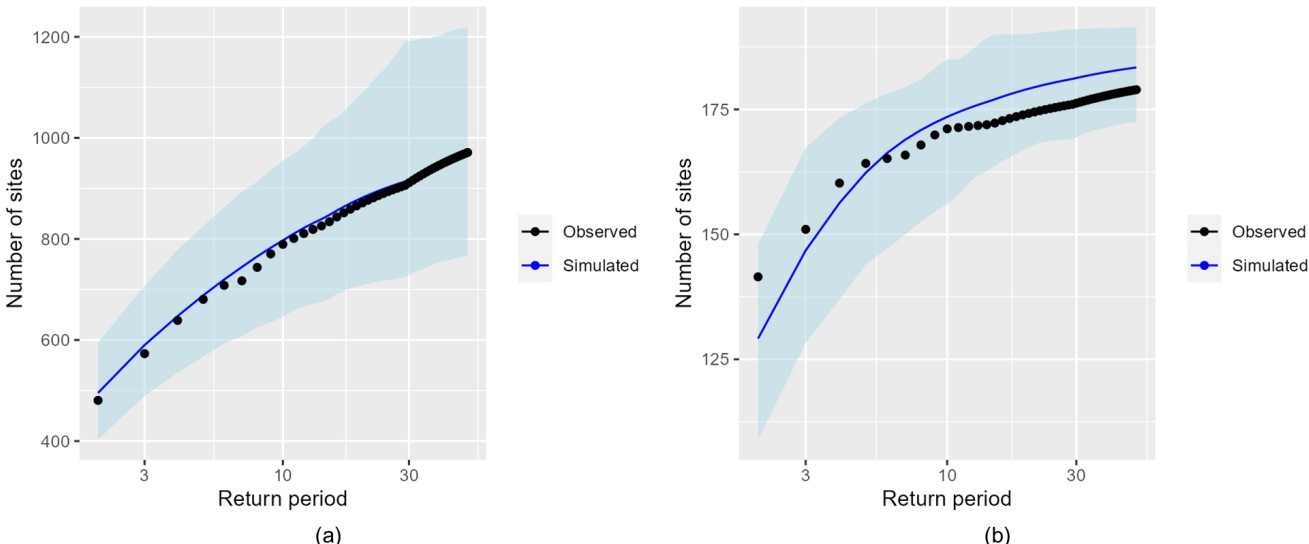

**Figure 9.** Annual loss exceedance curves calculated with the observed (black dots) and simulated (blue line) event sets, for the annual aggregate loss (a) and annual maximum loss (b). The 95% confidence intervals for the simulated loss are included (blue ribbon). They are calculated from 10000 random 30-year samples from the simulated years.

To validate the allocation of events into synthetic years, we calculate for each year (in the observed and simulated event set) the sum of all event footprint sizes and the maximum event size. This is roughly equivalent to computing the annual aggregate loss (AAL) and annual maximum loss (AML), where loss is approximated by the number of gauge sites exceeding the $0.9^{th}$ quantile flow (Keef et al., 2013). Loss exceedance curves for the annual aggregate and maximum loss are then calculated by empirical estimation and compared for the observed (black dots) and simulated event set (blue line) in Fig. 9. We use 10000

random samples of 30 year period taken from the simulated years to calculate the mean and 95 % confidence interval for the simulated loss exceedance curves. This is meant to reproduce the uncertainty range stemming from having a limited record length of 30 years, as is the case with the observed event set. For both aggregate and maximum annual loss, the observed losses lie well inside the uncertainty range calculated with the simulated events. The estimates closely agree for the AAL, while the simulated AML is slightly lower than the observed AML for lower return periods (< 5 years). Overall, the annual loss statistics

are well reproduced by the simulated events.

The marginal back-transformation is validated by comparing the return period levels calculated from the events and those provided by the marginal model in section 4.3.2, taken as ground truth. This would ensure the validity of the GEV quantiles $q_{GEV}$ calculated in section 4.3.2, used to get back the extreme flow magnitudes. For the simulated events, return period levels cannot be calculated from annual maxima because some synthetic years have the maximum flow lower than the 0 quantile on

the GEV scale, so the actual maximum flows for those years are not known. Instead, for each gauge station the exceedances over the $0.99^{th}$ quantile threshold (on the event scale) are back-transformed to discharge values using the marginal model in





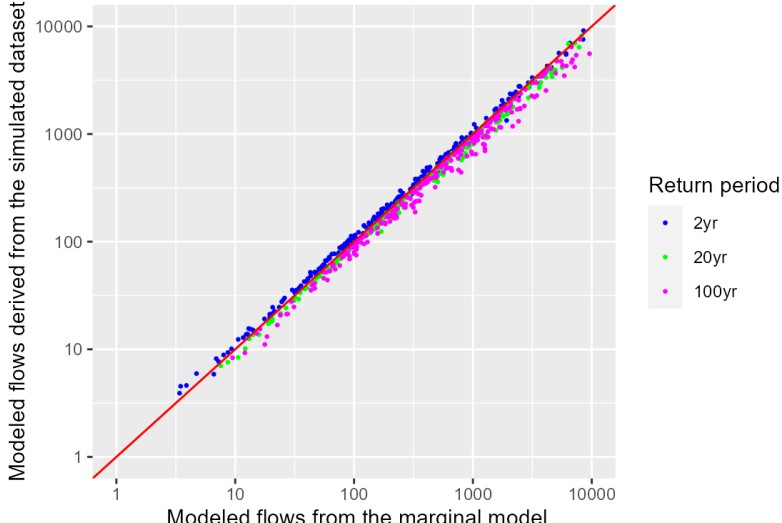

**Figure 10.** Discharge levels with return period 2 (blue), 20 (green) and 100 years (magenta) for all gauge stations, estimated from the synthetic event set and compared with the same quantities derived from the marginal model in section 4.3.2. The red line indicates the 1:1 fit.

section 4.3.2. They are then used to fit a Generalized Pareto distribution and derive the corresponding return period levels. Figure 10 shows that the return period flows calculated from the simulated events slightly overestimate the 2–year floods, and slightly underestimate the 20 and 100–year floods. Globally, the flows calculated from the simulated events are very close to the return period flows indicated by the marginal model in section 4.3.2.

# 6 Results

Events from the simulated event set are first inspected visually by plotting maps of some event footprints. In this section, the footprint of an event is defined as the number of catchments experiencing a flow greater than the 5–year return period flow. This will allow easier back-to-back comparisons with results from Quinn et al. (2019). Figure 11 shows four simulated events, chosen to cover a variety of spatial patterns contained in the observed records. The red dots represent all catchments experiencing a flow greater than the 1 in 5 year flow during a given event. Simulated events are diverse in location, footprint size and pattern. Each event is composed of clearly visible spatial clusters, suggesting that the strong spatial dependence of nearby extreme flows is well captured by the model.

Next, similarly to Quinn et al. (2019), we try to assess the link between event peak magnitude and footprint size. We expect that higher magnitude floods also present a larger footprint, as this is the conclusion drawn from Quinn et al. (2019)'s study for river floods in the US. For any event, its peak magnitude and epicenter are defined as the maximum flow experienced during that event and the location of that flow. Then for a given catchment, the mean event footprint size is defined as the mean footprint size of all events where that catchment is the epicenter. The mean modeled footprint size is spatially represented in Fig. 12 for





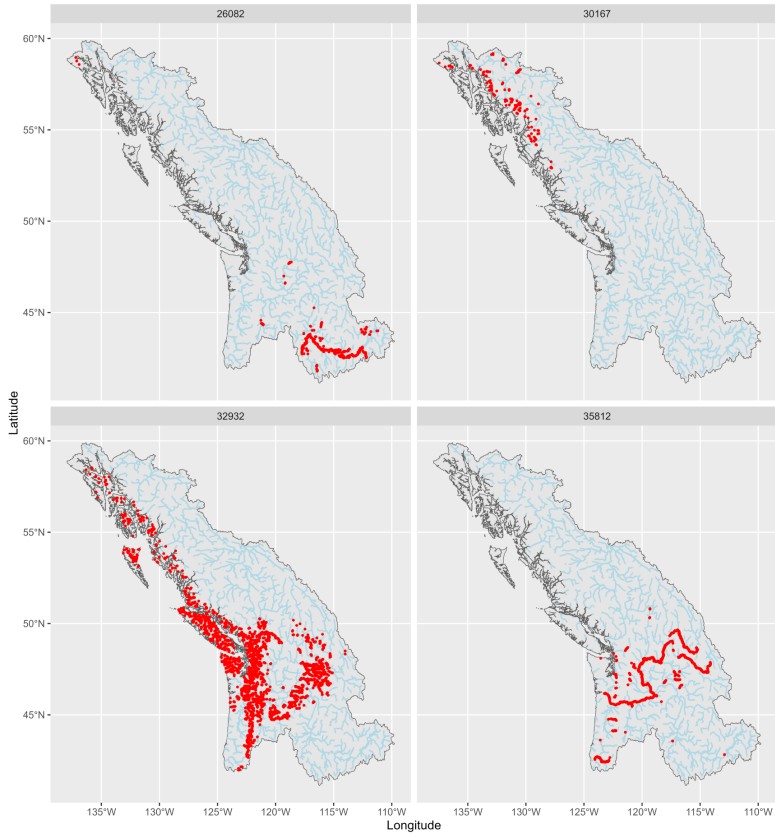

**Figure 11.** Examples of simulated events and their footprints. Red dots are catchments impacted by an event with a flow greater than the 1 in 5 year flow.

all catchments, where we have partitioned the events into four categories according to their peak magnitudes. These are: greater than 5, 5 to 20, 20 to 100 and greater than 100 year return period ranges, representing all, high, medium, and low frequency events. This analysis is meant to reproduce figure 9 in Quinn et al. (2019) to further validate the coherence of our simulated events against a reference paper in spatial flood modeling, except that here we calculate and show the mean event footprint size (in number of catchments) instead of the mean impacted area (in km$^2$). The plots show a positive correlation between event peak magnitude and footprint size. Higher magnitude events typically have a larger spatial footprint, though some spatial variability to this relationship does exist. For example, locations closer to the coastline (west side of the region) present a relatively low mean footprint size which increases less with event magnitude than inland locations. This might be partly explained by the geography of the coastline composed of many little islands and river networks, which restricts widespread flooding.

Finally, still following Quinn et al. (2019), we look at the distribution of flow magnitudes within each simulated event. To that end, the proportion $\pi$ of catchments experiencing a flow with return period greater than half the return period of the peak



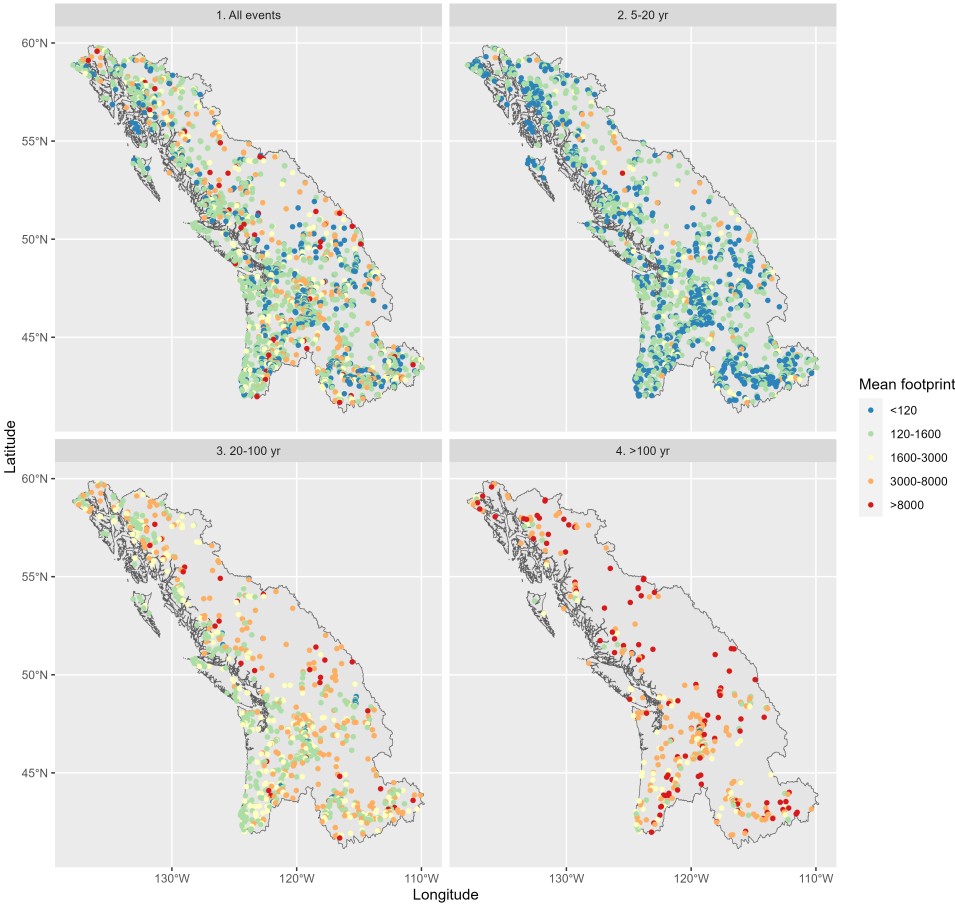

**Figure 12.** Mean event footprint size for catchments experiencing events with peak magnitude within one of four categories (> 5 (all events), 5–20, 20–100 and >100 year magnitudes). Each point corresponds to a catchment which is the epicenter of at least one event.

magnitude is calculated for each event:

$$\pi = \frac{\text{number of catchments with flow} > \text{half the return period of event magnitude}}{\text{number of catchments impacted by the event (flow with return period} > 5 \text{ yr})}$$

For example, for an event with a peak magnitude of 1 in 20 year flow, this proportion is equal to the number of catchments with at least 1 in 10 year flows divided by the number of catchments with at least 1 in 5 year flows. This is meant to represent the proportion of the event footprint which is relatively large. To distinguish the effect of event magnitude, histograms of the proportion of catchments with relatively large flows (as defined above) are plotted, where events are again divided into four categories of magnitude: greater than 5, 5 to 20, 20 to 100 and greater than 100 year return period (Fig. 13). For floods in the 5 to 20 year return period range, notice the peak at $\pi = 1$. This corresponds to all events with peak magnitude lower than the 1 in 10 year flow (by definition, all catchments impacted by such an event experience a magnitude greater than the 1 in 5 year flow). The plots show that as event peak magnitude increases, the proportion of the event footprint which is relatively large





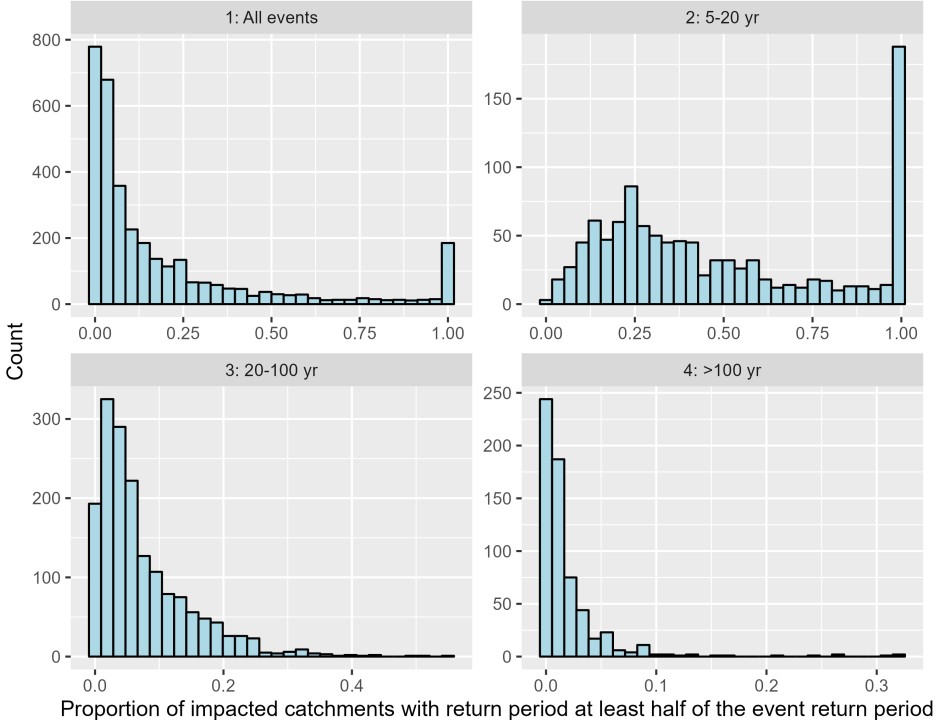

**Figure 13.** Histograms of the proportion of catchments in a given event with flow return period at least half the maximum event flow return period. The events are divided into four categories of magnitude: greater than 5, 5 to 20, 20 to 100 and greater than 100-year return period ranges.

decreases. The mean value of this proportion is 19 % for all events, 48 % for 5–20 year return period events, 7.4 % for 20–100 year return period events and 2 % for >100 year return period events. This means that for high magnitude events, despite their widespread behaviour (large event footprint size as shown in Fig. 12), significant flooding is concentrated in a small region within the event footprint (low proportion of the event footprint experiencing equivalently extreme flows). This finding agrees with the conclusions from Quinn et al. (2019).

# 7 Discussion

## 7.1 Data

The model we develop for flood spatial dependence is a statistical model which draws information from observed discharge records to simulate synthetic flood events. Station data does not have a homogeneous coverage over North America, with sparse gauge density in northern Canadian territories and California. However, most areas with less stations are also less populated, which mitigates the negative effects of model underperformance there. Preprocessing and the creation of the synchronous





regional datasets impose further temporal constraints which restrict the amount of available data used to fit the dependence
model. Any statistical model is only as good as the quality and quantity of information contained in the data used for estimation.
We are fully aware of these limitations and propose some ideas for future work to make use of the most high quality data
available. First, the station gauge data can be supplemented with recently developed remote sensing or radar data, although this

would require more extensive preprocessing to put the available sources on the same temporal and spatial resolutions. For some
regions, finding a sufficiently long common time period with a decent spatial coverage is not possible, due to gauge sites having
different times of operation. A less stringent condition could be used to subset the data when creating the synchronous discharge
set, and possible missing values can be filled using any reasonable gap filling algorithm, although this kind of procedure can
smooth out extreme events and induce additional uncertainty in the model. Finally, hydrological model outputs can be used

to augment the observed discharge records, as in Olcese et al. (2022); Wing et al. (2020), although this would also introduce
uncertainties linked to parameterizing the hydrological model. Additional efforts would also be needed to select suitable input
data to the hydrological model and downscale the output discharge.

## 7.2    Dependence model

The Fisher copula used to model the spatial dependence assumes that discharge pairwise Kendall's $\tau$ coefficients are positive.

This is mostly the case in our analysis, although when the considered region is vast and gauge sites are faraway, some Kendall's
$\tau$ might be negative. This drawback can theoretically be corrected by replacing the negative values with zero, although this
artificial correction induces a distortion on the entries of the correlation matrix $\Sigma$ after Kendall's $\tau$ inversion. A possible alter-
native to alleviate this difficulty is to define clusters of stations within a large region and apply the Fisher copula independently
on each cluster, though this would impose strict limits on the event footprints which might be unrealistic. Since the effect on

the $\Sigma$ matrix induced by the negative Kendall's $\tau$ is not overwhelming for our data, we did not pursue further modeling efforts
to explicitly account for negative Kendall's $\tau$.

The marginal modeling approach we follow is quite unconventional. Usually in a multivariate setting with copulas, the
variables are transformed to and back-transformed from the uniform scale using the same fitted distributions. We instead
make the choice to back-transform the simulated values using a marginal model operating on the annual scale, which fits

GEV distributions to annual maxima. This choice is mainly motivated by the better quality and quantity of annual maximum
discharge data at our disposal, compared to the regional event set. For example, the regional event set for region 9 spans 30
years and covers a subset of 253 gauge sites. In contrast, for this region, high quality annual maxima data are available for more
than 2000 stations, with records spanning 34 years in average. A dense gauge coverage over the region brings much needed
information to better predict extreme flows at ungauged catchments. Back-transforming values using a marginal model on the

annual scale requires transforming the simulated values from the event scale to the annual scale, a step which is validated
in section 5.3. The adopted approach does not allow recovering discharge values lower than the lower bound of the GEV
distributions (quantile 0 on the annual scale), but this is not a nuisance since we are interested in the extreme discharges only.

Using XGBoost to interpolate the Kendall's $\tau$ to ungauged catchments presents major advantages and improvement upon
previous methods used in the literature to model spatial patterns for ungauged locations. A machine learning model like



XGBoost draws its strength on the relatively high amount of information contained in the 130 covariates and the pairwise Kendall's $\tau$. Furthermore, using a decision tree based model allows to capture possibly non-linear patterns in the spatial structure of discharge. Finally, compared to more parameterized models like neural networks, boosting models like XGBoost are much faster to train and yield satisfying results.

The novel conditional simulation strategy for ungauged catchments is particularly suited to our data thanks to its compu-
tational efficiency. The British Columbia region is divided into 98980 catchments. The alternative of simulating values at all catchments at once would entail calculating a full correlation matrix of size $98980 \times 98980$. This is a huge computational challenge both in term of memory storage and calculation time, as such a matrix is not sparse and inverting it is virtually impossible. Moreover, in order to derive the full matrix, Kendall's $\tau$ between all pairs of ungauged catchments would need to be predicted. Those can be harder to estimate, especially for pairs of catchments which are geographically distant from any
gauge site. Finally, extending the correlation matrix to include all ungauged catchments would possibly alter the optimal degree of freedom $\nu$ of the Fisher copula, as the matrix size would be multiplied by the order of hundreds. In contrast, conditional simulation of each ungauged catchment changes the correlation matrix by only one column and row. Brunner et al. (2019) showed that the degree of freedom parameter of the Fisher copula is only weakly sensitive to an extension of the correlation matrix. Thus, since we only add one row and column, it is reasonable to assume that the estimated $\nu$ stays at the same value. In
summary, the conditional simulation strategy bypasses many complications which can arise if all simulated events were computed simultaneously. It avoids the prediction of Kendall's $\tau$ between ungauged catchments, the inversion of a huge correlation matrix and the recalculation of the degree of freedom $\nu$. The dependence model developed for ungauged catchments is thus computationally very efficient. When parallelized on 20 cores, 50000 simulated events for all catchments in region 9 can be computed in under half an hour.

**8   Conclusions**

This study introduces a novel methodology to model spatial dependence between riverine floods with application to North America, based on the Fisher copula (Favre et al., 2018; Brunner et al., 2019). The model developed in Brunner et al. (2019) is modified to be applicable to a much larger spatial scale and number of gauge sites. In particular, the Kendall's $\tau$ interpolation model uses a machine learning approach to draw strength from the 130 catchment attributes, and a conditional simulation strat-
egy is devised to simulate flood values efficiently, without computing the full correlation matrix. Using this model, a synthetic event set of arbitrary size can be simulated, with characteristics resembling the observed events and a more accurate sampling of the most extreme events. As this study aims to highlight the novelty of the introduced methodology and its validation, for results we focus our attention on the general picture drawn in the simulated events rather than possible local details. We find that the similar general conclusions as in Quinn et al. (2019) can be drawn with regard to flood spatial dependence, namely
that:

- Riverine floods usually occur in spatial clusters, reflecting the strong connectivity of river networks.

- Higher magnitude flood events tend to have a higher event footprint.



- Despite being widespread, higher magnitude flood events tend to have the most extreme magnitudes concentrated around a small area.

The spatially coherent event set can be integrated in a hydraulic model and flood loss model to calculate flood depth, damage to buildings and financial losses for each simulated event. Because spatial dependence is explicitly modeled, we expect that losses calculated from the event catalog are more reliable. Overall, the methodology presented in this study provides a valuable tool for stakeholders (governments, municipalities, insurance and reinsurance companies) to better understand and manage flood risk, contributing to enhance society's global resilience to flood disasters.

*Data availability.*  The gauge data used in the paper are available online in the GRDC database (https://portal.grdc.bafg.de/) , on the USGS website ( https://waterdata. usgs.gov/nwis), on dedicated Canadian and Quebec platforms (https://collaboration.cmc.ec.gc.ca/cmc/hydrometrics/www/ and https://www.cehq.gouv.qc.ca/atlas-hydroclimatique/index.htm). The catchment covariates are compiled from data sources available online. The output event set and model data layers are proprietary of Geosapiens Inc. in nature but can be made available for research use only. For further inquiries regarding access to our data and code, please contact Geosapiens Inc. directly.

*Author contributions.*  The methodology, coding, statistical analysis, result validation and manuscript writing were undertaken by Duy Anh Alexandre. Guidance, manuscript reviewing and some coding were undertaken by Chiranjib Chaudhuri. Catchment delineation procedures and some manuscript writing were undertaken by Jasmin Gill-Fortin. All authors have read and agreed to the published version of the manuscript.

*Competing interests.*  The authors declare no conflicts of interest.

*Acknowledgements.*  This research was partly funded by the National Research Council of Canada Industrial Research Assistance Program. We'd like to thank fellow Geosapiens team members Mingke Erin Li and Amit Kumar for their help with downloading and providing information on used datasets. We also thank former team member Fan Zhang for her help in the delineation of the HydroBASINS regions.



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
