# Peer review of "Novel extensions to the Fisher copula to model flood spatial dependence over North America"

_EGUsphere, 2024_

## Author Response (AR1)

**Novel extensions to the Fisher copula to model flood spatial dependence over North America**

Duy Anh Alexandre, Chiranjib Chaudhuri, and Jasmin Gill-Fortin

Responses to referee #1

- **One of the innovations of this study is the use of a machine learning model to predict the Kendall's τ instead of the original regression model. What are the advantages of choosing XGBoost for prediction instead of a regression model? Is there any comparison of results to provide evidence? Or is there any relevant material to illustrate this?**

XGBoost differs from linear models like simple regressions because it allows for nonlinearity through using decision trees as the base learner. This is particularly suitable for predictions involving hydrological processes which are highly non-linear. It is also an ensemble learner (combining many weak learners), which usually outperforms a single more complicated model. For example, for region 9 the Kendall's τ cross-validated RMSE is 0.079 for our XGBoost model and 0.171 for a linear regression model. To further justify our choice of model, a recent study shows that among state-of-the-art machine learning models, tree-based models still outperform more complex deep learning models on tabular data [1].

- **Line 207-208: "Since we work with stations spanning a large spatial scale, some pairs of stations present a negative Kendall's τ (15 % of the pairs for region British Columbia). Those values are replaced by zero before Kendall's τ inversion, since it requires the τ coefficients to be positive." The value of Kendall's τ is supposed to be negative, however, the authors have artificially converted it to zero. Does this practice have an effect on the results of the later calculations? More details should be furnished please.**

The inference of the Fisher copula parameters relies on the 1-to-1 relationship to link empirical Kendall's τ between pairs of observations and the corresponding entry of the correlation matrix $\Sigma$. However, this relationship is only monotonous on the range of Kendall's τ going from 0 to 1, as illustrated in figure 1 from Favre et al. (2018) [2]. Therefore, negative observed Kendall's τ are replaced by zero before Kendall's τ inversion. For these pairs of stations, this is equivalent to assuming that there is in reality no discharge correlation and the (small) observed negative correlation is spurious. In region 9, 15 % of the pairs present negative Kendall's τ but only 5 % have Kendall's τ lower than -0.2. For other regions, this percentage is even smaller.

This modification does result in a correlation matrix $\Sigma$ which is not positive definite, which requires an adjustment to make it positive definite, following the idea of Higham (2002). We assessed the deformation of $\Sigma$ by comparing each of its entries before and after the adjustment. This is shown in figure 1 for region 9. We see that the overall effects of transforming negative Kendall's τ to zero on $\Sigma$ are:
- a slight dampening of the correlations above 0.4
- a larger distortion of the zero entries, adjusted to values in the range [-0.2, 0.2]

Overall, the mean absolute difference is 0.062 between the adjusted entries and the original entries, which is deemed acceptable for our modeling purposes. As demonstrated in the manuscript, the simulated events are able to reproduce patterns in the observed events, in term of spatial patterns and upper tail dependence.

[Figure]

Figure 1: Entries of the Fisher copula correlation matrix Σ
before (x-axis) and after (y-axis) adjustment, for region 9

In conclusion, the practice of replacing negative observed Kendall's τ by zero does slightly modify the corresponding Fisher copula correlation matrix, but this is unlikely to have a negative impact on the quality and ability of the simulated events to reproduce characteristics of observed events.

**• P.8 -The Fisher copula is adopted for model the spatial dependence of riverine flood. What are the advantages of adopting this method over others in this case, such as regular copula, vine copula? Please add detailed elaboration.**
The Fisher copula is suitable for modeling spatial dependence in the right tail of random variables because:
-        It is non-symmetrical, allowing for asymmetry in the dependence pattern for the lower and upper tail (unlike the Gaussian or Student-t copula)
-        It allows to model positive upper tail dependence (unlike the gaussian copula)
-        The specification of pairwise dependence strength through a correlation matrix allows spatial interpolation to model ungauged catchments
-        It has a fairly low number of parameters, which make the model setup and inference much simpler than a vine copula for example. This is particularly important for operational reasons, as we are applying the model to thousands of catchments simultaneously.

Table 2 from [3] compares and summarizes the different characteristics of various copula models, including the Fisher copula.

• **P.11- The Generalized Extreme Value (GEV) distribution is utilized as the marginal distribution as annual maximum discharge in all catchments. Why is the GEV distribution function chosen directly? Was it selected preferably after comparison with other distribution functions?**

Historically, several different distributions have regularly been utilized to model annual maxima in climate and hydrological studies, including the Gumbel, GEV, Gamma, Log-Pearson type III [4]. However, the GEV distribution is more and more asserted as a common choice for modeling block extremes, with studies showing its better performance and goodness-of-fit compared to the Gumbel, Log-Pearson III or Log-normal distributions. For river discharge, examples of such comparative studies are [5-7]. Also, the GEV distribution arises as the asymptotic limit distribution for block maxima, ground in extreme value theory [8]. This allows better confidence in the ability of the GEV distribution to extrapolate values in the upper tail, where observations are typically rare or absent. Due to its flexible parameterization, the GEV distribution is able to capture a wide variety of right-tail behaviours (bounded, exponential decay or heavy-detailed) and we believe that its theoretical support and wide use in the hydrological community make it a suitable choice in our study.

• **P.5- This study omit analysis on regions number 1, 2, 3, 4, 13 and 14. The methodology and results are presented for region 9–British Columbia. And the results of regions 7 (St Lawrence), 8 (Prairie), 10 (East Coast) and 11 (Midwest) can be found in the supplementary information. What is the reason for selecting regions 7, 8, 9, 10 and 11 from these 14 regions for analyzing? Is there anything unique about these regions? Why are the other regions omitted? And what about regions 5, 6, and 12? They are not mentioned in the paper.**

The North American continent was divided into 14 regions following the level 2 HydroBASINS product delimitation. Among those, results for regions 7 to 11 were presented in our work (region 9 in the manuscript and regions 7, 8, 10 and 11 in the supplementary information). The study's main objectives were to present and validate the methodology developed, contributing to enhance modeling of flood spatial dependence in North America. Therefore, the results were presented for a subset of regions as a way to validate the methodology on a variety of different hydrological and climatological conditions and were not meant to be exhaustive. Regions 7 to 11 were prioritized owing to their higher population densities and gauge density, which resulted in more accurate results. Besides, from an operational perspective for Geosapiens, these regions are more important because they cover the

totality of the major urban centers in Canada, where our derived product will first be deployed. Results are however also available for regions 1 to 6, although their quality is lessened, and some minor methodology changes are required to account for the absence of high-quality discharge data in these northern territories. We omitted analysis on regions 12 to 14 because:

- From an operational perspective, they are the regions with no overlap with the Canadian territory.

- They have considerable overlap with the Mexican territory, where quality data gathering is not currently undertaken.

**• Line 215: "In this way, the correlation matrix Σ is extended to include all catchments, and the new parameters are used to simulate discharge at all catchments." The tense of this sentence needs to be modified. Please consider changing to the past tense.**

This is modified in the manuscript.

**• Line 447-448: "Finally, compared to more parameterized models like neural networks, boosting models like XGBoost are much faster to train and yield satisfying results." Is this conclusion derived from the comparison of the results calculated in this study? Or is it a regular characterization of XGBoost derived from other studies? Please provide some explanation to support this conclusion.**

Neural network models were not tested in our study. Following the principle of parsimony, we first started testing results with less parameterized models, namely Bayesian ridge regression (not presented) and boosting models. Since the predictive power of XGBoost was deemed satisfying, we did not see the need to pursue testing other more parameterized models. Besides, it is commonly known that tree-based models are faster to train than heavily parameterized neural network models. This can be seen for example in table 10.1 from [9] which compares some characteristics of different machine learning methods. This table shows that decision trees are computationally more scalable than neural nets. A recent study [1] also finds that boosting models outperform neural nets on medium sized tabular datasets, notably because they are more robust to uninformative features, less sensitive to the orientation of the data, while maintaining a superior computational speed.

**• Please adjust the font size in the images to make it larger and clearer, such as Figure 5, Figure 6, Figure 11 and so on.**

This is adjusted for figures 1, 5, 6, 11 and 12 in the manuscript.

**References**

[1] Grinsztajn, L., Oyallon, E., & Varoquaux, G. (2022). Why do tree-based models still outperform deep learning on tabular data? https://doi.org/10.48550/arXiv.2207.08815

[2] Favre, A.-C., Quessy, J.-F., and Toupin, M.-H.: The new family of Fisher copulas to model upper tail dependence and radial asymmetry: Properties and application to high-dimensional rainfall data, Environmetrics, 29, e2494, https://doi.org/10.1002/env.2494, 2018.

[3] Brunner, M. I., Furrer, R., and Favre, A.-C.: Modeling the spatial dependence of floods using the Fisher copula, Hydrology and Earth System Sciences, 23, 107–124, https://doi.org/10.5194/hess-23-107-2019, 2019.

[4] Nerantzaki S. D., Papalexiou S. M., Assessing extremes in hydroclimatology: A review on probabilistic methods, Journal of Hydrology, Volume 605, 2022, 127302, ISSN 0022-1694, https://doi.org/10.1016/j.jhydrol.2021.127302.

[5] Haktanir, T., Horlacher, H.B., 1993. Evaluation of various distributions for flood frequency analysis. Hydrol. Sci. J. 38, 15–32. https://doi.org/10.1080/ 02626669309492637.

[6] Moisello, U., 2007. On the use of partial probability weighted moments in the analysis of hydrological extremes. Hydrol. Process. 21, 1265–1279. https://doi.org/10.1002/ hyp.6310.

[7] Gubareva, T.S., Gartsman, B.I., 2010. Estimating distribution parameters of extreme hydrometeorological characteristics by L-moments method. Water Resour. 37, 437–445. https://doi.org/10.1134/S0097807810040020.

[8] Coles, S., Bawa, J., Trenner, L., and Dorazio, P.: An introduction to statistical modeling of extreme values, vol. 208, Springer, 2001.

[9] Hastie, T., Tibshirani, R., and Friedman, J.: The Elements of Statistical Learning, Springer, 2009.

1. **Are all the gauging stations are located in natural river basin? What if the gauged flood data is affected by reservoir operation? How to consider the influence of human intervention in floods.**

   In our study, we did not make an explicit distinction between stations in natural river basin and affected by reservoir operation. Nonetheless, during our preprocessing step to select high-quality stations, we tested the riverflow stationarity with a Mann-Kendall test and discarded stations where the trend is considered significant. This has a side effect of eliminating stations where human intervention can cause a disruptive change in the discharge time series. Furthermore, among the 130 static covariates used to predict fluvial flooding in our model, two are linked to human intervention and may capture the specific effect linked to stations with significant human intervention. They are percentage of urbanized land (LULC7) and the total upstream area protected by dams based on the GOODD dataset (log_dam_area). The GOODD dataset contains more than 38,000 dams as well as their associated catchments, allowing the analysis of their impacts in hydrological studies [1].

2. **The research mainly focus on the analyzing peak magnitude and total quantity of floods. Actually, the entire processes of flood hydrograph are worth more attention. Is it possible to show the simulation results over some specific flood events considering the entire time horizon of a flood.**

   Our study of fluvial flood spatial dependence focuses on the extreme floods and as such, the peak magnitude for the whole duration of each flood event is the main quantity of interest. As such, each flood event is summarized by its peak flow for every impacted location, so our simulated floods do not describe the whole time horizon of the flood. Other simulating components can be integrated to simulate the flood duration, hydrograph curvature, time of peak flow, etc... but we consider these developments to be outside the scope of the current study, which focuses on the spatial dependence of extreme fluvial floods (as summarised by the flood peak magnitude).

3. **How to verify the model results of the simulated flood footprint, as shown in Figure 11 d. Can the model compare the simulated footprint result with some actual floods?**

   The simulated flood footprint was validated using various aggregated metrics to be able to compare them to the observed floods. For example, the size of each event footprint (defined as the number of gauges impacted by a significant flow during a

flood) is validated against the observed event set (figure 7 and 8, manuscript). As such, it is not straightforward how to compare a simulated event directly with another historical flood event. However, for each historical flood event, we found that a simulated event with a similar footprint is present in the simulated event set (this matching is done using the F1 score to calculate similarity between event footprints). We present a non-exhaustive example of 4 historical floods with distinctive patterns, and their best matching simulated flood events.

[Figure]

**4. Is there any specific techniques in generating multi-site footprint to reduce computation effort? Would the computation of river-basin wide dependent flood cause trouble?**

The conditional simulation technique described in section 4.2 of the manuscript is precisely developed with the aim of generating multi-site footprint events in a computationally efficient way, without having to calculate and invert a complete correlation matrix for all unit catchments, which would be a substantial computational burden. Using this approach, for a given simulated flood event, realized flood values for all gauged stations are used to sequentially simulate values at each ungauged catchment. This computation step is very fast and when run on 20 CPUs in parallel on a personal computer, takes less than 10 minutes to be completed.

5. **390-395, what is number of catchments with flow > half the return period of event magnitude. The two paragraphs read confusing. Can you explain more over Figures 12 and 13.**

Figure 13 represents the proportion of catchments in a given event with flow return period higher than the flow corresponding to half the maximum flow return period (RP). This is more difficult to articulate in words than to understand. Suppose a flood event has a maximum flow (in RP scale) corresponding to RP100. Then we would count the number of catchments exceeding their respective RP50 flows (50 = 100/2), and divide by the number of catchments affected by the flood event (defined as having greater than RP5 flow). If another event had a maximum peak flow corresponding to RP20, we would count the number of catchments exceeding their respective RP10 flows (10 = 20/2), etc. The idea is to quantify for each flood event, the extent of the most impactful region. If this proportion is low, this means that the most extreme floods are localized at a few catchments, even if the flood event footprint can be large.

Figure 13 then plots the histogram of that proportion for all events (1), for events with maximum flow in the range RP5-RP20 (2), for events with maximum flow in the range RP20-RP100 (3), and for events with maximum flow greater than RP100 (4).

**References**

[1] Mulligan, M., van Soesbergen, A. & Sáenz, L. GOODD, a global dataset of more than 38,000 georeferenced dams. *Sci Data* **7**, 31 (2020). https://doi.org/10.1038/s41597-020-0362-5

---

## Author Response (AR2)

**Novel extensions to the Fisher copula to model flood spatial dependence over North America**

*Reply to reviewer*

1. Abstract highlights XGBoost as a key contribution for predicting discharge Kendall's τ coefficients, but the literature review lacks discussion or references to studies using XGBoost or similar machine learning models. It is suggested to include a review of relevant studies.

Studies using XGBoost or similar machine learning models are plentiful, but those which utilize machine learning methods to predict parameter values of a copula model are virtually inexistent. A reference is added in section 4.1 to a study showing that boosting models (especially XGBoost) are outperforming neural network on tabular data (line 223).

2. Methodology is complex, involving steps like the Fisher Copula, the XGBoost model, Kendall's τ interpolation, Conditional simulation, GEV parameter estimation, and Back-transformation. To help readers better understand the interrelationships and sequence of these steps, a flowchart is recommended.

Thank you for the suggestion. A flowchart has been included in the main article.

3. In Figure 9, the x-axis only shows the AAL and AML for return periods of 3, 10, and 30 years, while the focus of this study is on extreme discharge. It is suggested to explain why these specific return periods were chosen and whether observed and simulated data are available for longer return periods, such as 50 or 100 years.

The observed records have a limited length of 30 years, so the empirical return period levels can only be calculated up to return period 30 years.

4. In Figure 10, the discharge levels for return periods of 2, 20, and 100 years are all concentrated within 50-5000. It is suggested to explain why these discharge for different return periods appear to have similar ranges and to add units to the axes.

Some points corresponding to return period 100 years were left out in figure 10 due to the axis value limits. The axis value range has been corrected to include all points. The appearance of similar ranges is due to the logarithmic scale, but the leftmost points do

correspond to return period 2 and rightmost points to return period 100 years. Units have been added to the axes. Figure 10 has been updated.

5. It is suggested to explain the reason behind choosing the number of catchments impacted by the event with a return period flow greater than 5 years as the denominator in the formula (π) and whether different thresholds were considered during the analysis.

The number of catchments impacted by the event with a return period flow greater than 5 years is used to define an event's footprint size, as floods smaller than the 1 in 5-year return period are unlikely to lead to significant losses. This choice is also made in [1],  and we tried to use the same metric in order to compare our results with theirs. Different thresholds were not considered, as the event footprint size was defined that way earlier in the article.

6. In Fig. S3, it's puzzling why the same axis values (1e01, 1e01, 1e03, 1e05) are used across all subplots for return periods of 2, 5, 10, 50, 100, and 500 years. Typically, shorter return periods correspond to smaller discharges, while longer periods correspond to larger discharges.

The value range used to plot the discharges is wide enough to cover all the return periods from 2 to 500 years. A gradual shift toward higher values is visible as the return period gets higher, although this is small due to the logarithmic scale. The choice of the same axis values aims for a better homogeneity between the subplots.

References
[1] Quinn, N., Bates, P. D., Neal, J., Smith, A., Wing, O., Sampson, C., Smith, J., and Heffernan, J.: The Spatial Dependence of Flood Hazard and Risk in the United States, Water Resources Research, 55, 1890–1911, https://doi.org/10.1029/2018WR024205, 2019.